# Opportunities for Riboswitch Inhibition by Targeting Co-Transcriptional RNA Folding Events

**DOI:** 10.3390/ijms251910495

**Published:** 2024-09-29

**Authors:** Christine Stephen, Danea Palmer, Tatiana V. Mishanina

**Affiliations:** Department of Chemistry and Biochemistry, University of California San Diego, La Jolla, CA 92093, USAdepalmer@ucsd.edu (D.P.)

**Keywords:** bacteria, antibiotics, drug target, antisense oligonucleotide, riboswitch, riboswitch inhibitor, transcription, translation, RNA folding

## Abstract

Antibiotic resistance is a critical global health concern, causing millions of prolonged bacterial infections every year and straining our healthcare systems. Novel antibiotic strategies are essential to combating this health crisis and bacterial non-coding RNAs are promising targets for new antibiotics. In particular, a class of bacterial non-coding RNAs called riboswitches has attracted significant interest as antibiotic targets. Riboswitches reside in the 5′-untranslated region of an mRNA transcript and tune gene expression levels *in cis* by binding to a small-molecule ligand. Riboswitches often control expression of essential genes for bacterial survival, making riboswitch inhibitors an exciting prospect for new antibacterials. Synthetic ligand mimics have predominated the search for new riboswitch inhibitors, which are designed based on static structures of a riboswitch’s ligand-sensing aptamer domain or identified by screening a small-molecule library. However, many small-molecule inhibitors that bind an isolated riboswitch aptamer domain with high affinity in vitro lack potency *in vivo*. Importantly, riboswitches fold and respond to the ligand during active transcription *in vivo*. This co-transcriptional folding is often not considered during inhibitor design, and may explain the discrepancy between a low K_d_ in vitro and poor inhibition *in vivo*. In this review, we cover advances in riboswitch co-transcriptional folding and illustrate how intermediate structures can be targeted by antisense oligonucleotides—an exciting new strategy for riboswitch inhibitor design.

## 1. Introduction

Antibiotic resistance is a major threat to human health and causes millions of prolonged bacterial infections each year, risking overwhelming our healthcare systems [1]. Due to the widespread use of antibiotics to control infection, pathogenic bacteria have developed resistance to antibiotics that were previously effective, causing deaths from once-treatable infections [2]. Motivated by the pressing need for innovative antibiotic strategies, the World Health Organization (WHO) creates lists of priority bacterial pathogens to focus on when designing novel antibiotics [3]. One promising avenue for discovering new antibiotics is to target bacterial non-coding RNAs that control the expression of essential genes in pathogenic bacteria, such as riboswitches [4].

Riboswitches are 5′-untranslated regions of mRNA that typically respond to small-molecule ligands to control downstream gene expression via regulating transcription, translation, or RNA decay [5,6,7]. Importantly, with the exception of the thiamine pyrophosphate (TPP) riboswitch, riboswitches are only found in bacteria and archaea [8], making them promising targets for selective antibiotics with low side effects to the human host. Riboswitches are composed of two domains: an aptamer domain that binds a ligand, typically with high affinity, and an expression platform that undergoes structural changes to affect gene expression (Figure 1A,B) [9]. There are at least 55 classes of characterized riboswitches that are grouped by what ligand they bind, typically with a highly conserved aptamer domain [10]. Riboswitch expression platforms exhibit more sequence and structural diversity, and either promote or inhibit gene expression through ligand-dependent structural changes (Figure 1A,B) [5]. Expression platforms are designated as “off” or “on” depending on whether ligand binding prompts structural changes that downregulate (“off”) or upregulate (“on”) gene expression levels. “Off” riboswitch expression platforms form either an intrinsic terminator hairpin that prompts transcription termination (Figure 1A) or a hairpin stem that sequesters the ribosome binding site, leading to inhibition of translation (Figure 1B). Conversely, for “on” riboswitch expression platforms, ligand binding leads to the formation of an anti-terminator stem or release of the ribosome binding site, promoting downstream gene expression [5,11].

Riboswitches are typically found upstream of essential genes for metabolite biosynthesis and transport, thereby tuning expression of proteins that are essential for survival or virulence in some bacterial pathogens [12,13]. Since riboswitches are not found in human cells and often bind their ligands with high affinity in vitro (low nM K_d_ values) and high selectivity, the design of structural analogs that inhibit bacterial growth has attracted significant research interest [14]. Designing synthetic ligands to bind riboswitches with high affinity is the dominant focus of current research aiming to inhibit gene expression by targeting riboswitches (Figure 1C). These specific riboswitch inhibitors have been extensively reviewed in several recent papers [4,14,15,16].

To be an effective antibiotic that targets a riboswitch, compounds need to have high affinity for the riboswitch RNA, lack off-target effects on other metabolic pathways, have potential to avoid the bacteria developing antibiotic resistance, and pose low toxicity to human cells [15]. Importantly, many compounds that have high affinity for a riboswitch in vitro will not end up affecting gene expression in vivo, leading some researchers to turn to alternative methods, such as RNA-based gene silencing, to target riboswitches and inhibit bacterial gene expression (Figure 1D,E) [17,18]. Riboswitches fold co-transcriptionally and ligands often bind an early RNA folding intermediate, leading to the rearrangement of the expression platform as it is emerging from RNA polymerase (RNAP) [19]. Therefore, an inhibitor compound that has high affinity for the aptamer domain in isolation may lack potency in vivo if the inhibitor never reaches thermodynamic equilibrium with the riboswitch target, leading to a discrepancy whereby a low K_d_ does not indicate an effective inhibitor [17,20]. An exciting new strategy for synthetic RNA silencing of riboswitches, antisense oligonucleotides (ASOs), could potentially bridge this gap [18]. The aim of this review is to illustrate how ASOs can be a new tool for the development of antibiotics that target riboswitches, helping to overcome challenging cases where ligand mimics are unsuccessful. In the following sections, we first review the limitations of riboswitch small-molecule inhibitors, then shift the discussion to RNA-based gene silencing, covering how this occurs in nature and how such discoveries have inspired the design of synthetic oligonucleotides as a therapeutic strategy. From there, we discuss investigations into ASOs currently being researched to inhibit riboswitch-controlled gene expression in pathogenic bacteria. Lastly, we lay the foundation for how advances in riboswitch co-transcriptional folding can be leveraged to expand the available sequence space for ASO inhibitors targeting riboswitches.

## 2. Limitations in Designing Synthetic Ligands That Target Riboswitches in Pathogenic Bacteria

Despite some success in designing small-molecule riboswitch inhibitors, we introduce here some key riboswitches that are present in pathogenic bacteria for which significant limitations exist in the context of designing small-molecule inhibitors. Further, these riboswitches are excellent candidates for novel inhibition strategies, since they control expression of essential genes in bacteria. In Section 5, we continue the discussion of these riboswitches by highlighting how co-transcriptional folding studies can inspire the design of novel ASO inhibitors. This section is not intended to be a comprehensive review of small-molecule inhibitors targeting riboswitches present in pathogenic bacteria. For recent extensive reviews focused on small-molecule inhibitors, readers are directed to the following references [4,14,15,16].

### 2.1. TPP Riboswitch

The thiamine pyrophosphate (TPP) riboswitch is one of the most widespread riboswitches, found in a variety of pathogenic bacteria [21]. Thiamine is a critical cofactor for carbohydrate metabolism, and its biosynthetic pathway requires the expression of many unique proteins [22,23]. The TPP riboswitch is the only known riboswitch to occur in eukaryotic cells, with examples in plants and fungi [24]. The TPP aptamer, or THI-box, is highly conserved, and is made up of five stems that compact stably upon TPP binding [25,26]. The TPP riboswitch expression platform differs between bacterial species, and can either promote transcription termination or prevent translation initiation upon TPP binding the aptamer domain [27]. For example, in *E. coli*, there are multiple important TPP riboswitches that control downstream gene expression at both the transcriptional and translational level [28]. To highlight this diversity, the *E. coli thiC* riboswitch regulates the expression of proteins directly involved in TPP biosynthesis, while the *tpbA* riboswitch regulates the expression of an ABC transporter necessary to import thiamine and TPP into the bacterial cell [22,23].

A synthetic analog of thiamine (pyrithiamine, or PT) binds the thiamine pyrophosphate (TPP) riboswitch with nearly identical affinity to the natural TPP ligand [29]. PT was first characterized as an inhibitor of TPP-dependent enzyme production in *E. coli* in 1967, but mutant *E. coli* strains that resist PT inhibition by derepressing TPP-dependent enzyme synthesis by an unknown mechanism were isolated by 1969 [30,31]. In 2005, it was demonstrated that bacteria develop antibiotic resistance to PT through mutations to the TPP riboswitch, with mutations occurring in both the aptamer domain and expression platform [29]. Another study characterized binding by thiamine, methylene diphospohoric acid (MDP, a pyrophosphate analog), and a covalently linked version of the two molecules (TPPc) to the TPP riboswitch [32]. The individual thiamine and MDP bound with low affinity, but the conjugate molecule exhibited 6-fold higher affinity than TPP itself. The thiamine moiety of TPPc binds the riboswitch in a different conformation than the thiamine moiety of TPP, making it an interesting candidate to test in vivo [32]. Other studies have identified limited candidate inhibitors that bind the TPP riboswitch with sufficient affinity *in vitro*, but those have not been studied extensively in vivo [15].

### 2.2. FMN Riboswitch

The flavin mononucleotide (FMN) riboswitch class is widely distributed across bacteria and is found upstream of a variety of genes involved in flavin synthesis and transport [27]. Flavin-based cofactors are essential for proteins involved in the electron transport processes [33]. Since FMN is an essential cofactor for bacterial survival, inhibiting the flavin synthesis or transport pathways has attracted interest for antibiotic development. The FMN riboswitch aptamer domain has a six-stem junction that folds into a butterfly-like scaffold by identically folding the peripheral domains [34]. Additionally, the FMN riboswitch aptamer is largely folded in the absence of a ligand, and FMN binding only leads to local conformational changes [35]. These smaller-scale conformational changes ultimately alter the fold of the expression platform, leading to transcription termination or inhibition of translation initiation. For example, in *B. subtilis*, the *ribD* riboswitch promotes transcription termination upon FMN binding, leading to a downregulation of several FMN biosynthetic genes located in a single operon controlled by the riboswitch [36].

The FMN riboswitch can successfully be targeted by roseoflavin, which is phosphorylated by cellular proteins, but this requirement limits its efficiency [37]. Additionally, roseoflavin binds many off-target flavoproteins that are also present in humans, which gives it limited potential as an antibiotic [38,39]. Another ligand-mimic, ribocil, targets the FMN riboswitch aptamer with high affinity [40]. However, ribocil-resistant *E. coli* mutants have been identified, and importantly, all mutations conferring resistance mapped to the riboswitch aptamer domain or expression platform [40,41].

### 2.3. glmS Ribozyme

The *glmS* ribozyme is a self-cleaving ribozyme that responds to glucosamine-6-phosphate (GlcN6P) and is located upstream of GlcN6P synthetase [42]. GlcN6P synthetase is responsible for producing GlcN6P [42], which is the essential first step in the pathway to building a bacterial cell wall, so inhibiting this production prevents bacterial growth [43]. GlcN6P binding initiates the rapid self-cleavage of the *glmS* ribozyme, causing the release of a 25-nucleotide (nt) leader mRNA sequence that promotes subsequent degradation of the mRNA by RNase J1 [43]. The *glmS* ribozyme has a compact structure containing multiple pseudoknots that is largely unchanged by ligand binding [44]. Instead, GlcN6P participates directly in catalyzing the self-cleavage event [43,44].

Thousands of ligand analogs for the *glmS* ribozyme have been identified and screened, but very few were found to be active in promoting ribozyme self-cleavage. Of these, none activated the ribozyme as strongly as the natural ligand, making them less promising antibiotics [45,46,47].

### 2.4. Purine Riboswitch

Purine riboswitches respond to purine molecules and include guanine and adenine riboswitches [48]. These riboswitches occur upstream of genes involved in essential purine metabolism pathways, purine salvage pathways, and purine transport [48,49]. Thus, designing inhibitors for these essential pathways would lead to a loss of bacterial virulence [50,51]. Purine riboswitches have conserved aptamer domains consisting of three helices that nearly fully surround the three-way junction that coordinates the ligand [52]. In fact, the only difference between the adenine and guanine riboswitch aptamer binding pockets is one nucleobase that directly base-pairs with the ligand [53].

Purine riboswitches can be successfully targeted by rationally designed small molecules [54]. However, these compounds often show limited affinity for the riboswitch aptamer [49]. The compounds that are identified to have high affinity often show little to no effect on gene expression [17,20]. One example study generated 16 rationally designed small molecules and identified 6-*N*-hydroxylaminopurine (6-*N*-HAP) as a promising guanine analog that binds the guanine riboswitch aptamer with high affinity in vitro [49]. Unfortunately, when added to media, 6-*N*-HAP demonstrated low potency, since a high concentration was required to inhibit bacterial growth. Additionally, other designed compounds in this study inhibited growth, but exhibited off-target effects [49], and 6-*N*-HAP has even been demonstrated to be a general and potent mutagen in bacterial and mammalian cells [55,56]. These limitations to using small-molecule ligand mimics to inhibit riboswitch-controlled gene expression have led some researchers to explore RNA-based silencing methods, which will be reviewed in the next sections.

## 3. RNA-Based Silencing in Natural Contexts

RNA-based gene silencing (RNA interference, or RNAi) was described in eukaryotes in 1998, when double-stranded RNAs were identified as post-transcriptional gene silencing elements in *C. elegans* [57,58]. RNAi is a conserved eukaryotic mechanism that regulates gene expression at the transcriptional or post-transcriptional level [59]. RNA silencing relies on double-stranded RNA (dsRNA) cleavage that results in small RNAs [60,61]. These small RNAs are then recruited to guide protein complexes to silence the target genes [62]. This silencing process is a key mechanism by which eukaryotes defend against viruses and transposable elements [63], but is also used to regulate endogenous genes within eukaryotic cells [64].

In prokaryotes, small non-coding RNAs (sRNAs) are responsible for diverse gene silencing events [65]. The expression of sRNAs allows for the fine-tuning of gene expression in response to various environmental signals [66]. These sRNAs can prevent translation of genes by binding and occluding ribosome binding sites, or by affecting RNA stability and promoting degradation by RNase enzymes [67,68]. Some sRNAs associate with chaperone proteins, such as Hfq or ProQ, to achieve this translation inhibition and RNase-dependent degradation [69,70]. Other sRNAs sequester CsrA-like RNA-binding proteins that affect mRNA stability and translation [71]. Additionally, sRNAs associate with Cas proteins to promote the degradation of DNA or RNA targets as part of the CRISPR/Cas defense system against viral infection [65,72,73].

Diverse RNA-based gene silencing events are thus widespread among all kingdoms of life and allow for versatile and specific changes in gene expression in response to environmental signals. This versatility makes sRNAs a promising avenue for designing drugs to target and silence specific genes, especially in bacteria [57].

## 4. Synthetic RNA-Based Silencing

Drawing inspiration from natural RNA-based gene silencing, synthetic antisense oligonucleotides (ASOs) can be designed to achieve silencing of target genes. ASOs are single-stranded synthetic oligonucleotides that are complementary to a target gene [74]. ASOs can be used to silence specific genes by Watson–Crick base-pairing with target RNA transcripts and inhibiting gene expression as a result [18]. For example, an ASO may inhibit translation by targeting the RNA transcript for cleavage (Figure 2A) or by creating a double-stranded region that serves as a steric block for translating ribosomes or for proper RNA folding (Figure 2B) [75]. The first ASO shown to inhibit gene expression in vivo was discovered in 1978 and targeted viral RNA translation [76]. Since then, the development of ASOs as potential drugs has been widely pursued, and over a dozen ASOs have been approved by the FDA and others are in clinical trial [77,78].

Studies in the Penchovsky lab have shown that ASOs can be designed to target specific riboswitches (Figure 1D), including SAM-I, *glmS*, FMN, and TPP riboswitches (Figure 1E) from pathogenic bacteria [77]. These riboswitch-targeting ASOs have been previously reviewed [27], and the main findings are summarized briefly below.

All four riboswitch-targeting synthetic DNA ASOs were designed to target a portion of the conserved riboswitch aptamer domains [27]. These ASOs enabled cleavage of the riboswitch RNA by RNase H to degrade the RNA and prevent downstream gene expression (Figure 1D). For each riboswitch, the ASO was designed to be fully complementary to a 13–16 nt single-stranded sequence of riboswitch RNA that does not self-hybridize. For all four ASOs, the average minimum concentration of ASO inhibitor needed to inhibit 80% of bacterial growth (MIC_80_) was around 700 nM [27]. To avoid toxicity of the ASO, the sequences were chosen to be dissimilar to sequences occurring in human cells, and the ASOs were shown to be non-toxic at their MIC_80_ to a lung cancer human cell line (A549) [27]. A few different methods were used to demonstrate that the ASO was specific for a particular riboswitch. For the TPP and SAM-I riboswitches, the ASO was shown to be ineffective in targeting *E. coli*, which was expected as *E. coli* lacks these target riboswitches [21,79]. For the FMN riboswitch, a second ASO was designed with eight mismatches with the target RNA to be used as a negative control, and this mismatched ASO was ineffective at binding the FMN riboswitch [80]. Finally, the *glmS* ribozyme ASO was designed to be fully complementary to the aptamer domain of a target *glmS* ribozyme in *S. aureus*. This ASO did not inhibit the growth of *E. coli* or *B. subtilis*, demonstrating the desired specificity for the *glmS* ribozyme in *S. aureus* [81].

The advances in ASO-directed degradation of riboswitch RNAs show that riboswitches are a promising target for new antibacterials [27]. However, the requirements that the target RNA be unable to self-hybridize, single-stranded, and absent from human cells limit the options for ASO sequences that can be designed and tested for efficacy [27]. An exciting potential direction for antibacterial ASO design is to target single-stranded regions of RNA co-transcriptional folding intermediates to inhibit riboswitch activity and degrade the mRNA [82].

## 5. Advances in Riboswitch Co-Transcriptional Folding for Riboswitch Families Found in Pathogenic Bacteria

RNA folding is by nature a co-transcriptional process, since RNA secondary structural elements can form as fast as ~10^−6^ s [83]—multiple orders of magnitude faster than the transcription elongation rate by RNAPs, e.g., ~10^−3^ s/nt for *E. coli* RNAP [84]. Experimental evidence pointing towards the in vivo relevance of co-transcriptional RNA folding emerged from early studies on the *Tetrahymena* group I intron, where it was observed that the autocatalyzed splicing of this intron occurs in a few seconds [85] and thus must take place within the timeframe of transcription. In stark contrast, when the same *Tetrahymena* group I intron RNA is synthesized *in vitro*, denatured and refolded, the majority of the RNA becomes “trapped” in intermediate folds that rearrange into the native state on a slow timescale (~1–100 min) [86]. Therefore, developing a model of how an RNA acquires its functionally relevant folds requires incorporating the transcription machinery into experimental methods.

Many structural studies of riboswitch aptamer domains follow classic in vitro protocols: the RNA is transcribed with T7 RNAP, followed by denaturation and refolding for structural characterization [87,88]. While such work has provided seminal insights into the architecture of riboswitch aptamer domains [89], this does not recapitulate how RNA folds *in vivo*. A key example stems from initial efforts to crystallize the *B. subtilus yvrC* cobalamin riboswitch, where the traditional method of denaturing gel purification, refolding, and incubation with a cobalamin ligand did not yield the ligand-bound aptamer for X-ray crystallography [90]. These traditional in vitro methods used to study folded RNA differ from how an RNA folds in the cellular environment in a few key ways. First, RNA is transcribed 5′-3′, and local secondary structures can begin forming before the full-length transcript is synthesized. Rather than adhering to a single free-energy landscape wherein the full-length transcript folds all at once, for RNA, the folding free energy landscape changes as the transcript is lengthened by RNAP [91]. Second, the rate of RNAP transcription alters RNA’s folded structure by dictating the time window for forming structural motifs. In particular, transcriptional pausing—a temporary halt in nucleotide addition during transcript elongation [92]—creates a temporal window for key folding transitions and subsequent biological activity [93]. Lastly, the experimental conditions for studying RNA folding in a test tube do not recreate the local environment experienced by an RNA in the cell, which is a topic that has been extensively reviewed elsewhere [94].

To better recapitulate how riboswitches fold and respond to a ligand in a native context, more laboratories are incorporating the transcriptional machinery into their experimental methods. Structural approaches such as Selective 2′-Hydroxyl Acylation analyzed by Primer Extension and Mutational Profiling, or SHAPE-MaP [95], have been updated to incorporate RNAP, permitting the interrogation of riboswitch RNA structures in the context of transcription complexes [96]. Single-molecule approaches such as optical tweezers [97] and single-molecule FRET (smFRET) [98] have been used to study the folding events and dynamics of riboswitch constructs in the context of transcription complexes. Such studies have provided invaluable insights into structural intermediates traversed by riboswitch RNAs during active transcription that illustrate how a riboswitch aptamer domain “talks” to the expression platform [19,99]. In light of these advances, we posit that further consideration of riboswitch folding events during active transcription will present opportunities for designing novel inhibitors. We first review methods that hold significant potential to aid rational ASO design for targeting riboswitches co-transcriptionally: Co-transcriptional SHAPE-seq [100] and, more recently, variable length Transcription Elongation Complex RNA structure probing (TECprobe-VL) [19]. Further, we present specific examples of riboswitch co-transcriptional folding studies with the aim of highlighting the diversity of techniques that have uncovered novel insights into kinetic control of riboswitch folding. Table 1 presents all riboswitch classes found in pathogenic bacteria where specific family members have had their co-transcriptional folding characterized, with references listed for further reading. Additionally, the riboswitches we focus on in this section are poised as excellent targets for novel inhibition strategies, since they are all present in pathogenic bacteria and control the expression of essential genes for bacterial survival. Thus, upon repression of gene expression by a riboswitch inhibitor, the bacterial cell will be starved of an essential metabolite and die [12].

### 5.1. Co-Transcriptional SHAPE-Seq and TECprobe-VL

The Lucks and Strobel laboratories have been pioneers in their development of strategies to model RNA structure in the context of transcription elongation complexes. Initially, Strobel and colleagues reported on the co-transcriptional SHAPE-seq method [100], which involves the following steps: (1) Preparation of a DNA template encoding the RNA sequence of interest with biotinylated dNMPs randomly distributed throughout the template. Streptavidin binds at the randomly distributed biotinylation sites, creating roadblocks to RNAP movement across every position on the DNA template. (2) Reconstitution of transcription *in vitro,* resulting in the formation of stalled transcription elongation complexes that are exposing the nascent, structured RNA transcript. (3) Chemical probing of the displayed RNA with either a SHAPE reagent (e.g., benzoyl cyanide) or a base-specific reagent (e.g., dimethylsulfate) followed by RNA processing steps to read out where the modification sites are and inform structure models [100]. More recently, SHAPE-seq has been superseded by the TECprobe-VL method, which improved upon the co-transcriptional SHAPE-seq by simplifying the library preparation steps to make the method more broadly accessible [19]. The introduction of TECprobe-VL by Szyjka and Strobel studied the intermediate folds traversed early on in transcription of the *C. bacterium* ppGpp riboswitch, demonstrating regions of the RNA that are initially single-stranded and ultimately rearrange to be base-paired in the full-length folded riboswitch [19]. While the ppGpp riboswitch itself is not an attractive antibiotic candidate since it does not regulate the expression of an essential gene for bacterial survival [117], the accessibility of TECprobe-VL permits its application to any riboswitch sequence of interest.

### 5.2. TPP Riboswitches

#### 5.2.1. *E. coli thiC* Riboswitch

The Lafontaine lab applied a suite of complementary biochemical approaches to define how ligand binding to the TPP-sensing *E. coli thiC* riboswitch induces riboswitch structural rearrangements during active transcription [101]. Oligonucleotide hybridization/RNase H digestion assays were performed to compare ligand-dependent structural changes when TPP is added co-transcriptionally vs. post-transcriptionally. Measurements of ligand binding kinetics from these data demonstrated that co-transcriptional ligand binding is approximately four-fold more efficient compared to post-transcriptional ligand binding [101]. Further, in vitro transcription–translation assays elucidated that co-transcriptional TPP binding is required for the riboswitch to fold into a translationally inactive conformation [101]. These observations strongly suggest that ligand sensing and subsequent structural rearrangements are coupled to transcription [101].

This study identified three transcriptional pauses encountered by RNAP during synthesis of the riboswitch expression platform [101]. Motivated by the known importance of transcriptional pausing for RNA folding and gene regulation [92], the researchers explored the structure of nascent RNA transcripts at these pause sites using two complementary in vitro techniques that interrogate RNA structure—oligonucleotide hybridization/RNase H digestion assays and SHAPE [118]. These data showed that while the nascent RNA adjacent to the first two pauses was competent to sense TPP (Figure 3A), the riboswitch structure at the third pause was no longer able to sense TPP; this loss of sensitivity is due to the nucleation of a hairpin stem that disrupts the folded aptamer domain in favor of an alternative fold (Figure 3B) [101]. Taken together, this study defined a key time window for ligand-sensing and subsequent structural rearrangements that are “bookmarked” by transcriptional pauses; therefore, a rationally designed ASO that prevents the disruption of the folded aptamer domain within this window could bias the riboswitch towards the “off” state regardless of the intracellular TPP concentration (Figure 3C). Alternatively, an ASO containing DNA bases can anneal to a riboswitch folding intermediate and prompt RNase H degradation (Figure 3C).

#### 5.2.2. *E. coli tbpA* Riboswitch

In the case of the TPP-sensing *E. coli tbpA* riboswitch (a.k.a. *thiB* riboswitch), smFRET was used by the Lafontaine lab to monitor the dynamics of nascent riboswitch transcripts in the context of transcription complexes [104]. The *tbpA* riboswitch has a critical feature in common with the *thiC* riboswitch above, in that there are three transcriptional pauses during expression platform folding. Since prior work had found pausing to set a temporal window for ligand-sensing for the *thiC* riboswitch [101], this smFRET study investigated RNA dynamics at the *tbpA* pauses using roadblocked transcription complexes. Importantly, this study found that while nascent riboswitch transcripts were competent to bind the ligand at the first two pauses, ligand-induced structural changes were no longer observed at the third pause [104]. The analysis of riboswitch structural dynamics in this smFRET setup was expanded to monitor the folding of nascent riboswitch RNAs in real time, providing insight into TPP-induced folding transitions during active transcription [104]. These data reveal that TPP sensing occurs within a precise transcriptional window bound by the nucleation of a hairpin stem that competes with a paired region in the aptamer domain [104]. This insight mirrors the findings highlighted above for the *thiC* riboswitch, demonstrating the generality of an ASO-targeting approach that focuses on nucleating RNA structures in the vicinity of transcriptional pause sites (Figure 3C).

The Lucks lab reported new insights into intermediate folds traversed by the *E. coli tbpA* riboswitch. The co-transcriptional SHAPE-seq approach [100] was applied to nascent *tbpA* transcripts to define key positions during riboswitch folding that commit the RNA structure to either a translationally active or inactive conformation. These data reveal that the linker region between the aptamer domain and expression platform is involved in nucleating a hairpin stem. The nucleation of this hairpin stem requires breaking a key pairing interaction in the aptamer domain, and the completion of the hairpin folding results in RBS exposure [103]. Unlike the smFRET study highlighted above, the co-transcriptional SHAPE-seq experiments performed in this work map intermediate RNA folds to specific riboswitch transcript lengths. Therefore, analysis of the transcript lengths where competing RNA structures begin to fold will define specific sequence targets that, upon annealing of an ASO, could bias RNA folding towards the translationally inactive conformation, starving the bacterial cell of an essential cofactor. We present secondary structure models of these experimentally validated *tbpA* riboswitch folding intermediates in Figure 4 and highlight how these intermediates could be targeted by an ASO.

### 5.3. FMN Riboswitches

#### *B. subtilus ribD* Riboswitch

The Breaker laboratory clearly demonstrated that FMN-sensing by the *ribD* riboswitch in *B. subtilus* is kinetically driven, wherein ligand binding must occur within a precise time window to prompt transcription termination [105]. Specifically, transcriptional pauses were found to facilitate coordination between ligand binding and expression platform folding [105], similar to the TPP riboswitch examples highlighted above. Further, this study ascertained that FMN never reaches thermodynamic equilibrium with the riboswitch aptamer due to the kinetic constraints on the folding regime [105], highlighting the importance of transcription kinetics for effective riboswitching and downstream gene regulation.

Notably, further efforts to identify FMN riboswitch structural intermediates have not yet been reported. Approaches described earlier in this section such as co-transcriptional SHAPE-seq [100], and more recently TECprobe-VL [19], will be useful strategies to precisely characterize intermediate RNA structures in the vicinity of transcriptional pause sites and provide new targets for ASO inhibitor design.

### 5.4. glmS Ribozyme

#### *B. subtilus glmS* Ribozyme

Recently, the Woodson lab explored whether the *glmS* ribozyme executes its self-cleavage co-transcriptionally [107]. Prior to the publication of this study, few key datapoints suggested that co-transcriptional self-cleavage was possible: (1) the ribozyme self-cleavage reaction is fast (80 min^−1^) [119], so it is a reasonable assumption that cleavage occurs before the 1803-nt *glmS* open reading frame (ORF) is transcribed; (2) co-transcriptional cleavage permits RNase J1 to promptly begin mRNA degradation and prevent ribosomes from initiating translation of the *glmS* mRNA. If cleavage were to occur after ORF transcription, initiating ribosomes may impede RNase J1 degradation [120,121,122].

Motivated by the above observations, the Woodson lab employed a combination of single-molecule fluorescence microscopy and biochemical methods to analyze co-transcriptional ribozyme cleavage [107]. First, in vitro transcription reactions with radiolabeled RNAs showed that the ribozyme was able to self-cleave during transcription [107]. However, this bulk ensemble approach could not define the sequence window where co-transcriptional cleavage began. To address this, a single-molecule TIRF (smTIRF) microscopy assay was developed to monitor transcription and ribozyme activity in real time. These data were fit to a quantitative model, which ultimately demonstrated that the nascent ribozyme can initiate cleavage before completion of its synthesis [107]. Armed with the knowledge that the *glmS* ribozyme induces self-cleavage co-transcriptionally, the modeling of structural intermediates traversed by this ribozyme could identify sequence regions amenable to ASO targeting.

### 5.5. Purine Riboswitches

#### 5.5.1. Adenine Riboswitches

The Block lab employed an optical-trapping assay to monitor co-transcriptional folding events for the *B. subtilus pbuE* adenine-sensing riboswitch [97]. These assays clearly demonstrated that the adenine-induced formation of an anti-terminator stem occurs co-transcriptionally [97], highlighting a recurring theme for riboswitches that control transcription termination/anti-termination: the RNA structural switch must occur within the time-frame of transcription [123]. Once an anti-terminator stem is formed, RNAP continues transcribing the mRNA, and any subsequent RNA structural rearrangements in the 5′-UTR can no longer exert control over transcription.

More recent work from the Batey laboratory employed a structure-guided mutagenesis approach to characterize specific regions of the *pbuE* riboswitch that confer ligand-dependent regulatory activity [110]. First, it was observed that a hairpin stem in the aptamer domain facilitates strand invasion by the expression platform to promote the formation of an intrinsic terminator hairpin [110]. Second, the introduction of a strong transcriptional pause towards the end of aptamer domain synthesis produced the greatest induction of reporter gene expression of all the riboswitch variants tested in this study [110]. This finding underscores the importance of pause sites for effective riboswitching, and provides new clues for the rational design of ASO inhibitors. For example, since pauses provide a temporal window for RNA structural elements to form [93], transient single-stranded regions of RNA in the vicinity of a pause site could be targeted by an ASO inhibitor, as illustrated for the TPP-sensing riboswitches described in Section 5.2 (Figure 3 and Figure 4).

#### 5.5.2. Guanine Riboswitches

The Lucks lab investigated the co-transcriptional folding mechanism of the *B. subtilus yxjA* guanine-sensing riboswitch, which is an example of a riboswitch where there are no overlapping sequences between the aptamer domain and expression platform [108], unlike the above *pbuE* adenine-sensing riboswitch example. Interestingly, the riboswitch relies on the formation of a central helix that dictates the switching mechanism depending on whether guanine is bound to the aptamer domain (Figure 5A) [108]. Binding of guanine effectively prevents this central helix from folding, biasing the downstream riboswitch folding toward the “off” state [108]. The use of co-transcriptional SHAPE-seq defined the precise transcriptional window where this key central helix forms (Figure 5B) and led the authors to the hypothesis that this central helix forms through a strand-invasion process that is disfavored in the presence of ligand [108]. This hypothesis was further tested using gene reporter fusion assays and MD simulations, supporting a model wherein the formation of this central helix competes with a stable pairing interaction in the aptamer domain [108]. These findings underscore the importance of considering a riboswitch’s intermediate folds in efforts to bias its structural ensemble to the “off” state using novel inhibitors. Since this work on the *yxjA* riboswitch has experimentally validated folding intermediates, we present some of the secondary structure models in Figure 5B,C to highlight how these intermediates could be targeted by an ASO. Specifically, we propose that a steric block ASO designed to mimic the central helix could promote the folding of the downstream terminator hairpin in the absence of guanine ligand.

### 5.6. Cobalamin, c-di-GMP, and preQ_1_ Riboswitches

While less widely distributed in pathogenic bacteria, the cobalamin, c-di-GMP, and preQ_1_ riboswitches present an enticing opportunity for narrow-spectrum inhibitors that target specific bacterial strains. The co-transcriptional folding of the *E. coli btub* cobalamin-sensing riboswitch was first characterized by the Pan lab in a study that used oligonucleotide hybridization/RNase H digestion assays. First off, comparison of cobalamin-dependent structural changes between denatured and refolded and co-transcriptionally folded RNA elucidated that ligand-dependent structural rearrangements occur during active transcription [112]. Further, transcriptional pausing was found to facilitate cobalamin-dependent structural rearrangements in the expression platform [112]. While the oligonucleotide hybridization/RNase H digestion assays performed in this work were not intended to be applied to riboswitch inhibitor design, a revisitation of these biochemical studies could inspire initial efforts towards ASO design. For example, since this study employed 57 different oligo probes, a fresh look at these data could provide useful starting points for ASOs that can anneal to the nascent riboswitch RNA and prompt degradation during active transcription.

The Schwalbe lab has employed NMR to study folding intermediates of the *C. dificile* c-di-GMP riboswitch. Using a method to rapidly produce and purify a library of intermediate RNA transcripts [124], this study measured K_d_ values for riboswitch folding intermediates with both NMR and ITC [114]. These data defined a 61-nt window for c-di-GMP sensing by the riboswitch, and the K_d_ values obtained for the riboswitch transcripts within this window spanned two orders of magnitude [114]. The wide range of K_d_ values was explained by the presence of competing riboswitch folds present at specific transcript lengths, altering the riboswitch structural ensemble and in turn ligand binding [114].

The Walter lab has employed both smFRET and cryo-EM to investigate the folding and dynamics of the *B. subtilus* preQ_1_ riboswitch. Investigation of the preQ_1_ riboswitch with smFRET uncovered that a transcriptional pause situated between the aptamer domain and expression platform is stabilized by a pseudoknot that directly contacts RNAP [115]. Further, it was observed that binding of the preQ_1_ ligand signals RNAP release from the pause, underscoring the coupling between riboswitch folding and transcriptional pausing [115]. Elaborating on this work, cryo-EM structures of paused elongation complexes containing the nascent preQ_1_ riboswitch RNA were solved in both the ligand-free and ligand-bound states [116]. This work revealed that in the absence of preQ_1_, the local riboswitch RNA structure “retracts” the RNA 3′ end within the RNAP active site, leading to a halt in nucleotide incorporation [116]. However, preQ_1_ binding to the riboswitch induces large-scale conformational changes in RNAP that reorient the RNA 3′ end at the active site and release the pause [116].

### 5.7. Other Advances in Riboswitch Co-Transcriptional Folding

In this section, we summarize additional riboswitch co-transcriptional folding studies that point to recurring themes in riboswitch-mediated gene regulation, which will likely prove useful for the design of ASO inhibitors.

There has been significant research effort directed towards the co-transcriptional folding of the ZTP riboswitch class, which is a signaling molecule produced in response to 10-formyltetrahydrofolate deficiency in the bacterial cell [125]. The Hua lab designed a single-molecule vectorial folding assay using an engineered superhelicase that sequentially releases fluorescently labeled riboswitch RNA at a speed comparable to that of bacterial RNAPs (~60 nt/s) [126]. This technique was applied to the *F. ulcerans* ZTP riboswitch, which elucidated that the riboswitch is kinetically controlled, since the rate of RNA release by the helicase altered ligand-dependent structural changes [126]. The folding of the *C. beijerinckii pfl* ZTP riboswitch has been extensively studied by the Lucks and Strobel laboratories using co-transcriptional SHAPE-seq, and more recently TECprobe-VL [19,99]. These studies uncovered the specific sequence and structural elements that modulate an internal RNA strand displacement, which is disfavored in the presence of a ligand [19,99]. Building upon this work on the *C. beijerinckii pfl* ZTP riboswitch, the Lucks lab discovered that altering the rate of strand displacement via mutagenesis tunes its dynamic range [127], and that altering the expression platform sequence and structure to slow RNA folding enhances riboswitch sensitivity [128]. Additionally, the Schwalbe lab has investigated the *T. carboxydivorans* ZTP riboswitch co-transcriptional folding with NMR to define the precise window of transcript lengths where the ligand can bind [129].

Another well-studied class is the fluoride-sensing riboswitch, which controls expression of a fluoride exporter to prevent fluoride from accumulating to toxic levels in the cytosol [130]. The co-transcriptional folding of the *B. cereus crcB* fluoride-sensing riboswitch has been characterized by the Walter lab using smFRET. These studies highlighted a dynamic interplay between the transcription factor NusA and the fluoride binding the co-transcriptionally folded RNA [131]. A further smFRET study on this riboswitch by the Walter lab characterized the dynamics of intermediate transcript lengths and established the role of Mg^2+^ in fluoride-mediated conformational changes [132]. Another study on this riboswitch from the Lucks lab employed a combination of in vitro transcription and cellular gene expression assays to demonstrate the role of pseudoknot base pairs on co-transcriptional structure switching [133]. There are additional recent works on riboswitch co-transcriptional folding for the manganese sensing riboswitch class [134], 2′-dG sensing riboswitch class [135,136], and the SAM-VI riboswitch class [137], which we direct the reader to for further reading on this topic.

## 6. Antisense Oligonucleotides That Act on RNA Co-Transcriptionally: A Case Study on a Group I Intron

Building off of prior work from the Turner laboratory [138], the Zhang laboratory reported the use of an ASO to target co-transcriptional self-splicing of the group I intron located within the 26S rRNA of *C. albicans*, a fungus [82]. First, in vitro transcription assays with radiolabeled RNA were used to monitor the efficacy of different ASO sequences to inhibit co-transcriptional splicing [82]. It was found that eight different ASOs targeting the P3-P7 core of the intron (Figure 6A) achieved half-maximal inhibition in the ~0.2–0.3 µM concentration range [82]. The addition of a 2′O-Me moiety to an ASO targeting the P3-P7 core (2′-O-Me 240-261R) enhanced splicing inhibition in vitro by an order of magnitude, exhibiting an IC_50_ of ~37 nM [82]. Prior work established that modifications at the ribose 2′ position enhance ASO annealing to its target by promoting a 3′-*endo* pucker conformation of the ribose, thus making it more RNA-like and increasing its propensity to base-pair with the RNA target [139,140]. Since the 2′-O-Me 240-261R ASO delivered promising results *in vitro*, it was used for the subsequent in vivo experiments described here. Confocal fluorescence microscopy confirmed the presence of the core-targeted ASO in the nucleus of *C. albicans,* and reverse-transcription PCR (RT-PCR) targeting the pre-rRNA confirmed ASO inhibition *in vivo*, with an IC_50_ of 2.30 µM [82]. The authors proposed that splicing inhibition would be lethal to *C. albicans* because if the intron-containing RNA is assembled into the ribosome, association with the small ribosomal subunit is blocked (Figure 6B) [141]. The authors demonstrated that the core-targeted ASO specifically inhibits the growth of an intron-containing *C. albicans* and was unable to inhibit growth for a *C. albicans* strain lacking the intron [82]. Further, the ASO was found to inhibit *C. albicans* growth in a dose-dependent manner, with an IC_50_ of 4.7 µM [82].

Taken together, these data illustrate that ASOs can be rationally designed to target RNA co-transcriptional folding. For the group I intron example highlighted above, the best-performing ASO for splicing inhibition targets a region of the RNA that is base-paired in its final folded structure, and therefore would not have been an obvious target resulting from the inspection of the fully folded intron (Figure 6A). Thus, assessing the final-folded RNA structure alone limits opportunities for the design of efficacious ASO inhibitors. In the remaining sections of this review, we cover advances made in the delivery of oligonucleotide therapeutics and progress in the use of ASOs in the clinic as antibacterials.

## 7. Delivery of ASOs

### 7.1. Chemical Modifications

Chemical modifications have proven to be one of the most effective strategies for enhancing oligonucleotide drug delivery. Modifications of the nucleic acid backbone, ribose sugar and nucleobase have been used extensively in the design of therapeutic ASOs [142]. Such modifications help improve the pharmacokinetics, pharmacodynamics, and biodistribution of ASOs [142]. A commonly employed backbone modification is phosphorothioate linkages (PS), where one of the non-bridging oxygen atoms of the inter-nucleotide phosphate group is replaced with sulfur [142]. PS linkages improve an ASO’s therapeutic efficacy by providing nuclease resistance and increasing its association with proteins in plasma and within cells [142]. Another common ASO modification is the addition of a functional group at the 2′ position of a ribose sugar, with common examples being 2′-*O*-methyl, 2′-*O*-methoxyethyl and 2′-fluoro [142]. These modifications increase resistance to nucleases by substituting the nucleophilic 2′-hydroxyl group, which improves oligonucleotide stability in plasma and increases its half-life in tissues, resulting in extended drug effects [142]. Lastly, rather than using a DNA or RNA-like oligonucleotide, alternative chemistries have been developed, such as phosphorodiamidate morpholino oligonucleotides (PMOs), which are charge-neutral nucleic acids where the five-membered ribose is replaced with a six-membered morpholine ring [142]. The advantage of charge-neutral oligonucleotide chemistries is that they facilitate conjugation to moieties that promote cellular delivery [142].

### 7.2. Bioconjugation Strategies

One strategy to deliver ASOs into bacteria is the use of cell-penetrating peptides (CPPs), which are short (generally <30 amino acids) amphipathic or cationic peptide fragments that originate from protein translocation motifs found in nature, or are based on polymers of basic amino acids [143]. Peptide–PMO (PPMO) conjugates consisting of a CPP covalently linked to a PMO have been identified as a promising antibacterial delivery strategy [142]. For example, PPMO conjugates that target an essential bacterial gene *acpP* in a murine infection model were demonstrated to be bactericidal [144]. A comprehensive list of PPMO conjugates that have been experimentally validated to cross the cell wall across diverse pathogenic bacteria is presented in a review by Xue et al. [145]. Additional strategies that are not based on the covalent modification of the ASO to the delivery system have attracted interest as alternatives, such as encapsulation in lipid nanoparticles or DNA nanostructures, and are also covered in the review by Xue et al. [145].

## 8. Translation of ASO Therapies to the Clinic

There has been significant progress in the translation of oligonucleotide therapies to the clinic, with thirteen ASO-based medicines to treat human diseases approved by the FDA thus far for clinical use [75]. For example, Golodirsen and Eteplirsen are FDA-approved ASOs used to treat Duchenne muscular dystrophy (DMD), which promote the skipping of an exon in the *DMD* gene that would otherwise produce an out-of-frame transcript [142]. Generally speaking, ASOs used to treat human disease can operate through many different mechanisms. For example, some are designed to silence disease-causing or disease-modifying genes by promoting RNase H degradation upon DNA oligo annealing to the target mRNA [142]. Other ASOs are designed to mask a particular stretch of sequence in an mRNA and interfere with downstream RNA–RNA and/or RNA–protein interactions, with a common application being the modulation of alternative splicing, as described for Golodirsen and Eteplirsen [142].

While there are no FDA-approved antisense antibacterials yet, there have been great strides in their development. Multiple preclinical studies delivered promising results on the use of ASOs as potential antibacterial therapeutics [146]. For example, an ASO targeting a gene in *S. aureus* that is critical for replication (*ftsZ*) inhibited the growth of methicillin-resistant *S. aureus* in cell culture, and conferred increased survival to mice infected with the Mu50 strain of *S. aureus* [147]. Additionally, in an *S. aureus*-infected cutaneous mouse wound, the use of an ASO targeting the essential *gyrA* mRNA decreased bacterial growth and promoted cutaneous wound healing in the mice [148]. Additional efforts to develop ASOs targeting essential genes in pathogenic bacteria are highlighted in the review by Buthelezi et al. [146].

Despite significant advances made in the design of ASO therapeutics, there are many challenges to overcome to achieve their widespread use in the clinic. The effective delivery of ASOs to many tissues is a major hurdle to overcome [142]. For example, injected nucleic acid therapeutics must resist nuclease degradation, bypass renal clearance, and avoid non-productive sequestration by plasma proteins in order to reach the desired target [142]. A combination of chemical modifications and conjugation strategies outlined in Section 7 is typically employed to confer the desired pharmacokinetic and pharmacodynamic properties to an ASO drug [142], and further advances on these fronts will likely expand the number of conditions that are treated with ASO therapeutics.

## 9. Discussion

Widespread antibiotic resistance in pathogenic bacteria is an urgent problem in modern medicine [1]. To combat this problem, riboswitches have emerged as potential targets for new antibiotics, since they are widespread among pathogenic bacteria but absent in human cells [4]. There have been significant advances in designing small-molecule riboswitch inhibitors, but these strategies still show significant limitations [29,37,49]. Antisense oligonucleotides are beginning to attract attention as a new riboswitch inhibition strategy, and several have been designed to specifically target riboswitches while being non-toxic to human cells [27]. However, there are limited studies leveraging advances in RNA co-transcriptional folding to design ASO riboswitch inhibitors. The group I intron example (Section 6) demonstrated that regions of RNA that are base-paired in the final folded structure can still be effectively targeted by an ASO. Advances in methodologies to model riboswitch folding intermediates [19,100] provide exciting opportunities to expand the search for efficacious ASO inhibitors.

Experimental work on the riboswitches outlined in Section 5 has built foundational principles for riboswitch co-transcriptional folding that can guide the search for novel ASO inhibitors. For example, co-transcriptional SHAPE-seq studies on the *tbpA* riboswitch illustrated how in the absence of the TPP ligand, a strand invasion event disrupts the riboswitch aptamer domain, leading to the riboswitch “on” state. We illustrate how a competitor ASO could prevent effective strand displacement of the riboswitch, biasing the structural ensemble towards the “off” state even in the absence of TPP ligand (Figure 2 and Figure 3). Additionally, biochemical study of TPP riboswitches can provide inspiration for novel ASO design, with an example being ASOs that target key RNA structural transitions in the vicinity of transcriptional pause sites during expression platform folding (Figure 2).

Additionally, there have been exciting advances in targeting dsRNA–ssRNA junctions using ASOs, which can form a triplex adjacent to a duplex at a dsRNA–ssRNA junction [149]. Recent work illustrated how targeting these junctions with an ASO can stimulate ribosomal frameshifting and inhibit Dicer cleavage activity of a pre-miRNA [149]. This work represents an important advance that could inspire the design of ASOs that target riboswitches at dsRNA–ssRNA junctions, further expanding the available sequence space for inhibitor design.

The studies referenced in Section 7 and Section 8 demonstrate how ASOs can be modified and/or conjugated to facilitate efficient delivery into the cell [142,143,145]. In fact, thirteen ASO-based drugs are already FDA-approved for clinical use, with more in various stages of clinical trial [75]. Using ASOs to treat infectious diseases, including bacterial infections, is an active area of research with several promising candidates already developed [146]. Taken together, we propose that the methods used to study RNA co-transcriptional folding could be key for designing novel ASOs to target riboswitch RNAs in pathogenic bacteria.

## Figures and Tables

**Figure 1 ijms-25-10495-f001:**
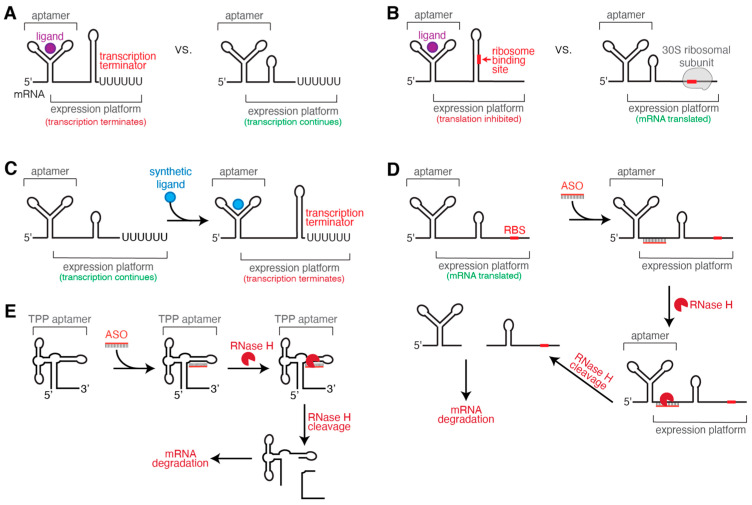
Overview of riboswitch-controlled gene expression and riboswitch inhibition strategies. (**A**) Riboswitch responding to ligand by terminating transcription, turning gene expression “off”. (**B**) Riboswitch responding to ligand by inhibiting translation, turning gene expression “off”. (**C**,**D**) Mechanism for inhibiting gene expression by targeting a riboswitch with a synthetic ligand analog (**C**) or a DNA-based antisense oligonucleotide (ASO) that marks mRNA for degradation via RNase H cleavage (**D**). (**E**) Example ASO-targeting of the TPP riboswitch aptamer domain, followed by RNase H cleavage and mRNA degradation.

**Figure 2 ijms-25-10495-f002:**
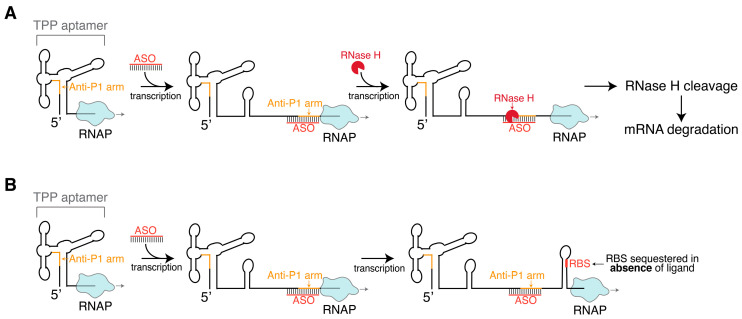
Overview of ASO-directed co-transcriptional silencing of target riboswitch-controlled genes, using the TPP riboswitch as an example. (**A**) ASO binding recruits RNase H, which cleaves the DNA-RNA hybrid, prompting mRNA degradation. (**B**) ASO binding creates a steric block that prevents a structural rearrangement and favors the formation of the RBS sequestering stem.

**Figure 3 ijms-25-10495-f003:**
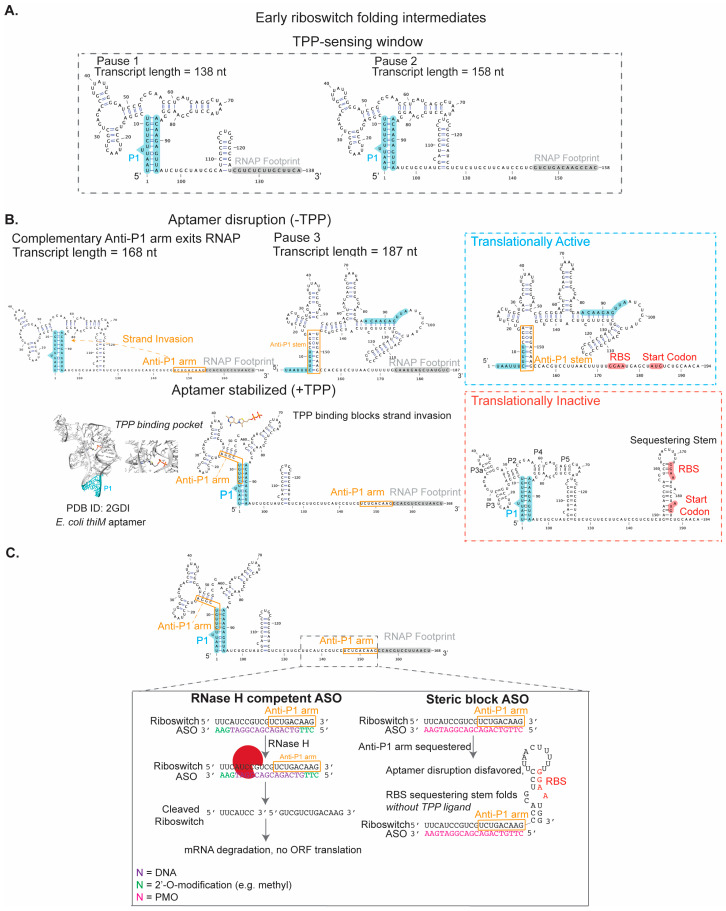
Folding pathway for *E. coli thiC* riboswitch in the presence and absence of TPP ligand and its proposed targeting with ASO. (**A**) The riboswitch structural intermediates in the vicinity of RNAP pause sites are modeled based on published structure-probing experiments. (**B**) In the absence of TPP ligand, the Anti-P1 arm invades the already-folded TPP-sensing aptamer domain, promoting a translationally active conformation. In the presence of TPP ligand, TPP stabilizes P1 in the aptamer domain, blocking strand invasion by the Anti-P1 arm and promoting a translationally inactive conformation. (**C**) Opportunities for ASO inhibition by sequestering the Anti-P1 arm and preventing its invasion of the folded aptamer domain. One strategy is an RNase H-competent ASO, which contains a complementary DNA strand flanked by 2′-OMe nucleotides at its 5′ and 3′ ends to enhance stability. Alternatively, an ASO can act as a steric block that remains stably base-paired to the riboswitch RNA. Since the ASO remains base-paired, strand invasion by the Anti-P1 arm is blocked, favoring the folding of a hairpin stem that sequesters the RBS. A phosphorodiamidate morpholino oligonucleotide (PMO) is a charge-neutral nucleic acid that is commonly used for steric block ASOs and is described further in Section 7.

**Figure 4 ijms-25-10495-f004:**
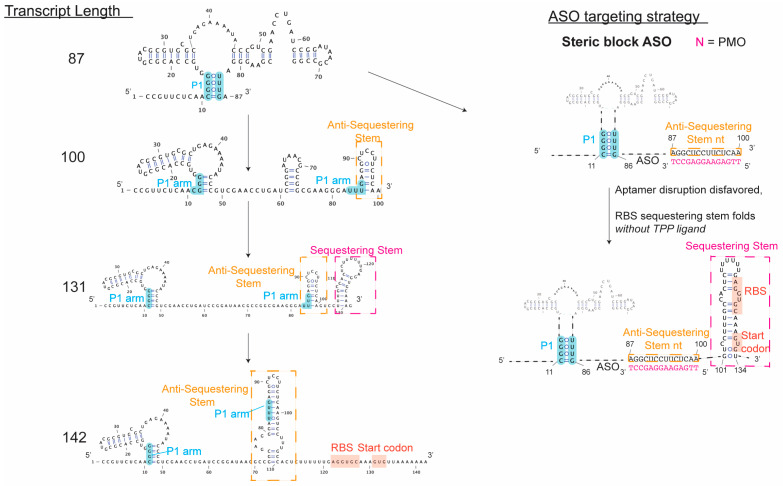
*E. coli tbpA* riboswitch folding intermediates experimentally validated by co-transcriptional SHAPE-seq. A steric block ASO (e.g., PMO) can prevent formation of the anti-sequestering stem and favor instead folding of the sequestering stem, leaving RBS occluded in the absence of TPP ligand.

**Figure 5 ijms-25-10495-f005:**
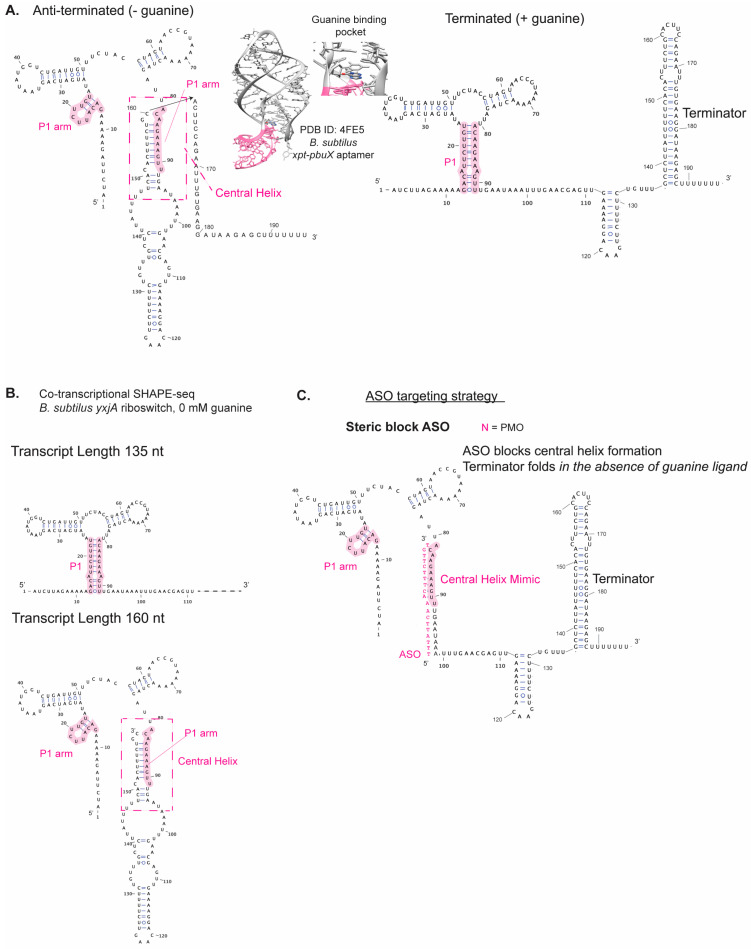
Folding outcomes for *B. subtilus yxjA* riboswitch in the presence and absence of guanine ligand and its proposed targeting with ASO. (**A**) Alternative structures of the *B. subtilus yxjA* riboswitch in the presence and absence of guanine ligand. (**B**) Riboswitch structural intermediates based on co-transcriptional SHAPE-seq experiments. (**C**) ASO targeting strategy to mimic the central helix, which can promote the folding of the downstream terminator in the absence of guanine ligand.

**Figure 6 ijms-25-10495-f006:**
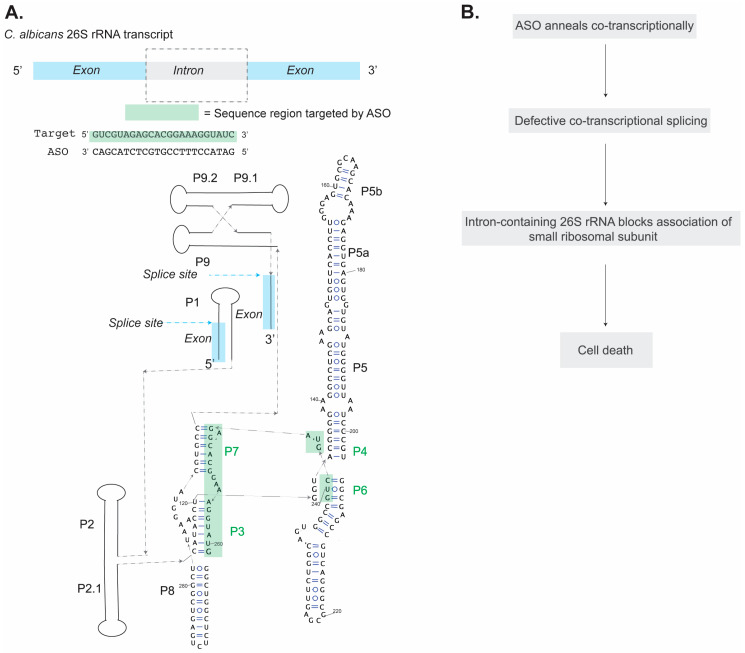
(**A**) Secondary structure of *C. albicans* 26S rRNA group I intron, with paired regions of the intron core targeted by the ASO labeled in green. The specific stretch of sequence targeted by the ASO is highlighted in green. Splice sites are denoted with blue arrows. (**B**) Illustration of how the ASO targeting co-transcriptional splicing of the 26S rRNA is lethal to *C. albicans*.

**Table 1 ijms-25-10495-t001:** Riboswitch classes in pathogenic bacteria whose co-transcriptional folding has been experimentally characterized.

Riboswitch Class	Occurrence in Human Pathogenic Bacteria	Co-Transcriptional Folding Studies	Experimentally Validated Secondary/Tertiary Structures for Folding Intermediates
**TPP**	Present in 48 species of human pathogenic bacteria [24]	*E. coli thiC* riboswitch: [101,102] *E. coli tbpA* (a.k.a. *thiB*) riboswitch: [103,104]	*E. coli thiC* riboswitch: [101,102] *E. coli tbpA* (a.k.a. *thiB*) riboswitch: [103]
**FMN**	Present in 41 species of human pathogenic bacteria [24]	*B. subtilus ribD* riboswitch: [105]	N/A
***glmS* ribozyme**	Present in 26 species of human pathogenic bacteria [24]	*B. subtilus glmS* ribozyme [106,107]	N/A
**Purine**	Present in 17 species of human pathogenic bacteria [24]	Guanine: *B. subtilus yxjA* and *xpt-pbuX* riboswitches: [108,109] Adenine: *B. subtilus pbuE* and *V. vulnificus add* riboswitches: [97,110,111]	Guanine: *B. subtilus yxjA* and *xpt-pbuX* riboswitches: [108,109] Adenine: N/A
**Cobalamin**	Present in 36 species of human pathogenic bacteria [24]	*E. coli btuB* riboswitch: [112]	N/A
**c-di-GMP**	Present in 7 species of human pathogenic bacteria [113]	*C. dificile* c-di-GMP riboswitch: [114]	*C. dificile* c-di-GMP riboswitch: [114]
**preQ_1_**	Present in 15 species of human pathogenic bacteria [113]	*B. subtilus preQ*_1_ riboswitch: [115,116]	*B. subtilus preQ*_1_ riboswitch: [116]

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
