# Peer review of "Opportunities for Riboswitch Inhibition by Targeting Co-Transcriptional RNA Folding Events"

_ijms, 2024, doi:10.3390/ijms251910495_

Round 1

Reviewer 1 Report

Comments and Suggestions for Authors

The review article by Mishanina and coworkers focuses on summarizing the strategy of using riboswitch cotranscriptional folding intermediate structures as antibacterial targets for ASOs. The article is timely and important in the field of novel antibiotics development as well as RNA folding and targeting field. The paper is well written and may be publishable after the authors address the below concerns.

Good to add cartoons (without showing the detailed sequences) for the co-transcriptional folding and targeting.

 Line 263 and Figure 5, cite a related paper:  https://www.pnas.org/doi/full/10.1073/pnas.172391199

Section 5.5A, Add a figure for adenine riboswitch folding and targeting.

Section 6, Add a brief discussion of a new strategy in targeting dynamic RNA structures (e.g., dsRNA-ssRNA junctions created by a translating ribosome):  https://doi.org/10.1016/j.xcrp.2024.102150

Reviewer 2 Report

Comments and Suggestions for Authors

This review article by Stephen et al. focuses on advances in development of synthetic antisense oligonucleotides (ASOs) as inhibitors targeting riboswitches. First, the authors discuss the limitation in designing small molecule inhibitors for targeting riboswitches. They underline existence of RNA-based gene silencing. Then, they thoroughly evaluate structure and co-transcriptional folding of different riboswitches, highlighting importance of targeting these features for ASO inhibitor design. In the end, they discuss strategies of ASOs delivery and application in the clinic.

Overall, the manuscript addresses important topic and provides new insights into the field. I would only suggest one minor addition. The authors extensively discuss in chapters 2 and 5 the top four riboswitch classes presented in Table 1; however, the three remaining riboswitches (Cobalamin, c-di-GAMP, and preQ1) from this table are not mentioned anywhere in the text. It would be consistent if the authors addressed in the text all riboswitch classes that they specified in the Table 1.   

Reviewer 3 Report

Comments and Suggestions for Authors

This review focuses on introducing the advances and opportunities for riboswitch inhibition, mainly discussing targeting co-transcriptional RNA folding events. The review also covers the understanding of riboswitch co-transcriptional folding and how this knowledge can be used to design ASOs that target riboswitch folding intermediates. The manuscript is well written, and the figures are well organized and clear. So, it is worthy of publication in IJMS. However, the following minor concerns should be addressed before publication.

1.     A lot of papers that the author cited are very old. Indeed, these works are important and significant. However, more recently published papers should be added to provide new perspectives in this field. For example, Breaker lab at Yale has done a lot of works on riboswitch study, but most of the citations are from early 2000s. Please add a few references within the past five years.

2.     The authors are highlighting ASO approaches for riboswitch. However, there are efforts made to target riboswitches by small molecules. Although some of the molecules show tight binding but low activity in cells. These are sill worth mentioning. For example, there are some synthetic ligands for PreQ1 riboswtiches (Nat. Commun., 2019,  10, 1501. https://doi.org/10.1038/s41467-019-09493-3) and ligands for TPP riboswitch (ACS Chem. Biol. 2022, 17, 2, 438–448. doi.org/10.1021/acschembio.1c00880).

3.     ASO approach for RNA targeting has its own advantages. Compared to small molecules, ASOs are more designable, with higher affinity and specificity. What is the challenge for ASO approach to be used as antibacterial drug? What efforts should be made to overcome the potential issues (such as delivery, and in vivo stability)?

4.     Please add more introduction about ZTP riboswitch, since it is also a good case of co-transcriptional RNA folding, and there are some reported studies by smFRET (Nat. Commun., 2020, 11, 4531. https://doi.org/10.1038/s41467-020-18283-1).

Comments on the Quality of English Language

English language is fine, only minor editing is needed.
